# Stereocomplex Polylactide for Drug Delivery and Biomedical Applications: A Review

**DOI:** 10.3390/molecules26102846

**Published:** 2021-05-11

**Authors:** Seung Hyuk Im, Dam Hyeok Im, Su Jeong Park, Justin Jihong Chung, Youngmee Jung, Soo Hyun Kim

**Affiliations:** 1NBIT, KU-KIST Graduate School of Converging Science and Technology, Korea University, 145 Anam-ro, Seongbuk-gu, Seoul 02841, Korea; ishidh@korea.ac.kr (S.H.I.); airplane96@kist.re.kr (S.J.P.); 2enoughU Inc., 114 Goryeodae-ro, Seongbuk-gu, Seoul 02856, Korea; 3Department of Mechanical Engineering, Graduate School, Korea University, 145, Anam-ro, Seongbuk-gu, Seoul 02841, Korea; idhish1@naver.com; 4Center for Biomaterials, Biomedical Research Institute, Korea Institute of Science and Technology (KIST), Seoul 02792, Korea; chungjj@kist.re.kr (J.J.C.); winnie97@kist.re.kr (Y.J.); 5School of Electrical and Electronic Engineering, Yonsei-KIST Convergence Research Institute, Yonsei University, Seoul 03722, Korea; 6Korea Institute of Science and Technology (KIST) Europe, Campus E 7.1, 66123 Saarbrueken, Germany

**Keywords:** polylactide, stereocomplex, biodegradable polymers, drug delivery system, biomedical applications

## Abstract

Polylactide (PLA) is among the most common biodegradable polymers, with applications in various fields, such as renewable and biomedical industries. PLA features poly(D-lactic acid) (PDLA) and poly(L-lactic acid) (PLLA) enantiomers, which form stereocomplex crystals through racemic blending. PLA emerged as a promising material owing to its sustainable, eco-friendly, and fully biodegradable properties. Nevertheless, PLA still has a low applicability for drug delivery as a carrier and scaffold. Stereocomplex PLA (sc-PLA) exhibits substantially improved mechanical and physical strength compared to the homopolymer, overcoming these limitations. Recently, numerous studies have reported the use of sc-PLA as a drug carrier through encapsulation of various drugs, proteins, and secondary molecules by various processes including micelle formation, self-assembly, emulsion, and inkjet printing. However, concerns such as low loading capacity, weak stability of hydrophilic contents, and non-sustainable release behavior remain. This review focuses on various strategies to overcome the current challenges of sc-PLA in drug delivery systems and biomedical applications in three critical fields, namely anti-cancer therapy, tissue engineering, and anti-microbial activity. Furthermore, the excellent potential of sc-PLA as a next-generation polymeric material is discussed.

## 1. Introduction

Over the past few decades, polylactide (PLA) has emerged as a common biomaterial in biomedical applications owing to its favorable properties, such as complete biodegradability, mechanical properties, biocompatibility, processability, and transparency [1,2]. A PLA possesses two types of three-dimensional helical structures that twist in clockwise (D-configured) and counter-clockwise (L-configured) directions. Since Ikada et al. first reported stereocomplex formation between enantiomeric PLA in 1987 [3], stereocomplexation between poly(D-lactide) (PDLA) and poly(L-lactide) (PLLA) enantiomers has been the subject of continuous study. This research has accelerated with the rapid growth of practical use and the potential worth of PLA as a representative biodegradable polymer. Stereocomplex crystallites with a 3/1 helical structure in the PLA material can overcome inferior mechanical and thermal characteristics of homo-crystallites having a 10/3 helical structure. The combination of two enantiomeric polymers increases the melting point (*Tm*) and crystallinity through the compact orientation of crystals in the material. Ultimately, this change in a biodegradable polymer can result in an increase in thermal stability, mechanical strength, resistance against solvent penetration, and external forces. Stereocomplex crystals in PLA have commonly been formed by solution blending, melt blending, emulsion blending, precipitation into non-solvent, and supercritical fluid (SCF) techniques [4,5,6,7,8,9]. Each method has different advantages and disadvantages regarding the yield and stereocomplexation efficiency, processability, solubility, time, and cost.

Recently, numerous studies have reported that stereocomplex polylactide (sc-PLA) with improved physical characteristics can be used in drug delivery and as molecular carriers [10,11,12]. Nanoparticles, such as microspheres and micelles of sc-PLA, have the advantage of controlling drug uptake and release patterns through their synthesis and modification, as well as natural adsorption. Despite these advantages, many challenges remain for the application of sc-PLA as a drug carrier, including inferior encapsulation efficiency, low stability of hydrophilic drugs and proteins, and the burst release phenomenon. To resolve these issues, various strategies, such as polymerization, self-assembly, surface modification, and polymer grafting, have been studied [13,14,15,16,17]. This review discusses various synthesis and processing methods for the application of sc-PLA as a drug and molecular carrier, and it suggests future directions to stimulate the application of sc-PLA with therapeutic molecules in drug delivery systems and biomedical applications.

## 2. Drug Delivery

### 2.1. Stereocomplexed Micelle System

Micelles exhibiting specific core–shell structures are widely applied in drug delivery systems, because they can load a variety of drugs owing to their good loading capacity. Furthermore, micelles have a higher thermodynamic stability than colloids under physiological conditions owing to the lower critical micelle concentration. Therefore, polymeric micelles induced by the self-assembly of amphiphilic block copolymers have been vigorously researched for their biomedical roles, such as target-specific carriers, nano-bioreactors, and non-viral gene vectors. However, polymeric micelles have limited applications because of a short circulation time, due to their rapid excretion via urine after intravenous injection, and difficulty in accumulation and providing sustainable drug release at the target site [18]. Strategies for chemical cross-linking of hydrophilic poly(ethylene glycol) (PEG) segments have been proposed since the 1990s to improve the stability of polymeric micelles. Gref et al. (1994) fabricated nanospheres composed of a core and shell formed by biodegradable polymers, such as poly(lactic co-glycolic acid) (PLGA), polycaprolactone (PCL), and their copolymers, covalently bonded with PEG [19]. The PEG coating could significantly increase the blood circulation time of carriers by reducing their detection and opsonization by macrophages in the reticuloendothelial system and decrease their accumulation in the liver. In addition, this injectable nanoparticle carrier could encapsulate up to 45% of its weight in a one-step procedure. Micelles induced by biodegradable polymers enable the control of drug release kinetics and time, as the degradation period varies with the molecular weight (*Mn*) of the polymer. Furthermore, this feature allows higher kinetic and thermodynamic stabilities than those of a surfactant micelle with a lower molecular weight [20]. Kang et al. reported that monodisperse stereocomplexed micelles could be obtained by self-assembly of the PLA-PEG block copolymer [21]. This report was the first to verify that PLA-based micelles can form stereocomplex configurations in aqueous conditions. The micelles exhibited improved kinetic stability, as well as both physical and chemical stabilities. In particular, secondary aggregation, which is known to be the main problem of conventional polymeric micelles, was reduced. Figure 1a shows an atomic force microscopy (AFM) image of a stereocomplexed micelle with a spherical shape of approximately 46 nm diameter and a narrow size distribution. Figure 1b indicates that the micelle has a normal X-ray diffraction (XRD) pattern and a small crystalline domain of sc-PLA.

### 2.2. Self-Assembled Nanoparticle

Stereocomplexes can build nanoparticles by self-assembly, apart from micelle formation. Non-covalent interactions, such as electrostatic interactions, hydrogen bonding, and hydrophobic-hydrophobic interactions are driving forces to induce self-assembly of amphiphilic block copolymers. The stability of the self-assembly can be influenced by several environmental factors, including the pH, temperature, polymer concentration, and ionic strength [22,23,24,25]. Bishara et al. synthesized stereocomplex particles composed of an enantiomeric triblock copolymer (PLA-PEG_2000_-PLA) by blending in acetonitrile solutions [26]. Figure 2a shows that the stereocomplex nanoparticles have smooth surfaces resulting from PEG segments and sizes ranging from the nanometer to micrometer range. The size of stereocomplex particles increased with increasing concentration of PLA in the triblock copolymer (Figure 2b). This size of biodegradable particles affects the drug release rate from a particular carrier. Smaller-sized stereocomplex particles have a faster degradation rate owing to their larger surface area, as shown in Figure 2c. In the in vitro release profile test, stereocomplex particles with triblock copolymer could encapsulate 80% of the water-soluble drug dexamethasone, which was completely released for 30 days (Figure 2d). Similar to the previous degradation results, particles with a higher PLA concentration could release drugs at a slower rate. Biodegradable particles were fully degraded during the two months after complete release of dexamethasone phosphate. The hydrophilic drug was assumed to accelerate the infiltration of water into the polymer material compared to the original polymer.

Liu et al. produced pH-sensitive stereocomplex nanoparticles consisting of methoxy poly(ethylene glycol)-poly(L-histidine)-polylactide (mPEG_45_-PH_30_-PLA_82_) tri-block copolymer by self-assembly [27] (Figure 3a). In this study, the mPEG_45_-PH_30_-PLLA_82_/mPEG_45_-PH_30_-PDLA_82_ stereocomplex stably maintained a mean diameter of 90 nm at pH 6.8, whereas the diameter of mPEG_45_-PH_30_-PLA_82_ increased to the micrometer scale under the same conditions. The mean diameter of the stereocomplex nanoparticles slightly decreased when the pH changed from 5.0 to 7.9, as shown in Figure 3b,c. It was considered that lower pH conditions caused swelling of the nanoparticles with protonation of poly(L-histidine) in the tri-block copolymer [28]. Transmission electron microscopy (TEM) revealed that the stereocomplex particles retained their spherical shape, ranging from pH 5.0 to 7.4, and the TEM image confirmed a reduction in the particle size at pH 7.9 (Figure 3d,e). Furthermore, the cell viability of the stereocomplex nanoparticles was 90 % higher than that of homopolymer nanoparticles in co-culture with mouse 3T3 fibroblasts. This was attributed to the reduction of cytotoxic PDLA segments through stereocomplexation. Numerous studies have demonstrated that stereocomplex particles are capable of loading not only proteins, but also drugs.

Lim and Park (2000) synthesized stereocomplex microspheres based on the solvent-casting method, after polymerization of PLLA-PEG-PLLA and PDLA-PEG-PDLA of a tri-block ABA copolymer [29]. Then, a bovine serum albumin (BSA) protein could be encapsulated in the stereocomplex by the double emulsion solvent evaporation method. Their study reported that stereocomplex microspheres showed a more sustainable and predictable release pattern of the protein than that of the homopolymer. This can be attributed to the hydrophilic PEG unit in the tri-block microspheres preventing aggregation and non-specific adsorption of the protein. In the in vitro release profiles of BSA, PEG stereocomplex microspheres and PEG tri-block copolymer microspheres exhibited a higher initial burst effect than that of PLLA microspheres, but the microspheres based on PEG showed a larger cumulative release (Figure 4). This is attributed to the water uptake capacity of the microspheres that increased owing to the PEG of the hydrophilic unit. After the burst release in the initial stage, the microspheres showed a relatively sustained release by a diffusion-controlled mechanism over 50 days.

### 2.3. Emulsion Blending

Generally, sc-PLA is prepared by mixing two enantiomeric polymers in solution or in a melted state. However, these methods result in low stereocomplexation efficiency, loss of molecular weights, and original properties. The layer-by-layer (LbL) technique based on a Pickering emulsion template has been commonly used to fabricate nano- and microcapsules [30,31,32,33,34]. Despite its facilitated application to various fields, this approach has several concerns, including the requirement of inorganic solid multi-layer precursors, linkers, and templates, and inferior integrity and loading efficiency of a capsule [35,36,37]. To overcome these limitations, emulsion blending based on the droplet-in-droplet method has emerged as an alternative method to fabricate polymeric micro-carriers. Brzeziński proposed a microfluidic approach based on a water-in-oil-in-water (W/O/W) double emulsion for the synthesis of hollow stereocomplex microcapsules [38]. In this study, 2-ureido-4[1H]-pyrimidinone (UPy)-functionalized PLA enantiomers formed stereocomplex microcapsules at the water-chloroform interface via a one-step microfluidic self-assembly (Figure 5a). The capsule could reversibly control the assembly and disassembly to the supramolecular functionality of the interfacial assembly, unlike other microcarriers. This enables the capsule to freely adjust the stiffness and permeability of its shell and drug release. Figure 5b shows the morphological and structural reorganization of the sc-PLA microparticles induced by the W/O/W double emulsion. In this observation, microdroplets with a mean diameter of approximately 260 µm appear to have a high monodispersity and narrow size distribution. In particular, microcapsules were divided from the water phase by spontaneous dewetting after the oil droplets shrank. The inner oil droplet could maintain hollow stereocomplexed microcapsules induced at the interface between water and chloroform [39,40,41]. Finally, the sc-PLA-UPy microcapsules were precipitated with a mean diameter of approximately 160 μm. Stereocomplex crystallites were assumed to act as efficient nucleating agents and interfacial enhancers during this reaction [42].

Im et al. developed a novel strategy for blending two homopolymers of PLA in an oil-in-water (O/W) emulsion state [43]. This O/W emulsion blending method facilitated the rapid combination of solutions of PLLA and PDLA enantiomers with the addition of an emulsifier and mechanical stirring in a one-pot reactor (Figure 6a) [43]. During blending, stereocomplex crystals could be formed simultaneously with the diffusion of oil-phased PLLA and PDLA into water, emulsification induced by an emulsifier, and mechanical mixing. Unlike other phases, the emulsion phase could induce significantly rapid stereocomplexation by promoting supramolecular interactions derived from lower interfacial tension. Moreover, this method can significantly improve the time and cost, availability, and efficiency compared to conventional methods. As a result, sc-PLA particles fabricated by emulsion blending showed spherical morphology with an improved stereocomplexation efficiency of up to 99% (Figure 6b,c). Furthermore, fluorouracil (5-FU), a cancer drug, can be encapsulated into sc-PLA particles for stereocomplexation during O/W emulsion blending, as shown in Figure 7a. In the drug release profiling test for 5-FU, sc-PLA particles induced by emulsion blending loaded 13 wt% of 5-FU, and the drug was slowly released for eight days after the initial burst of drug release (Figure 7b). To the best of our knowledge, this is the first report on an O/W emulsion blending method for simultaneously inducing stereocomplexation and drug encapsulation in an emulsion state.

### 2.4. Inkjet Printing

Recently, inkjet printing technology has been used in various industrial areas, including tissue engineering and organic or inorganic electronics. This technique can rapidly and exquisitely deposit printing materials, such as polymers, metals, nano- and micro-particles, and proteins and cells with highly controlled volume and patterning of link-droplets. Consequently, polymeric biomaterials can be LbL-assembled on substrates. Akagi et al. (2012) devised a fast printing method for sc-PLA based on LbL deposition using an inkjet printer without a redundant rinsing step [44]. Each PLLA and PDLA solution was separately printed at the same point, and stereocomplex crystals were then formed with enantiomeric homocrystallites during solvent evaporation. Figure 8 shows the schematics of the inkjet printing process for the LbL formation of sc-PLA. The first LbL-assembly method, employed by the authors for formation of the stereocomplex, involved the dissolution of PLLA and PDLA in chloroform, followed by the PLLA solution being sprayed onto a glass substrate. Then, the PDLA solution was printed over the substrate after drying the PLLA droplets (Figure 8a). One cycle indicates that PLLA and PDLA solutions were printed once on the substrate in turn. In the second LbL-assembly method, PLLA and PDLA were dissolved in chloroform, and the blended solution was printed on the substrate in the first step. Subsequently, the blended solution was printed over dried mixed droplets once again in the second step (Figure 8b). This method holds potential to provide rapid fabrication, as 1 × 10^5^ droplets were sprayed, and the processing time was approximately 100 s in each step. As shown in Figure 9a, the X-ray diffraction (XRD) patterns of PLA composites fabricated by the second method showed peaks at 12°, 21°, and 24° of 2θ degrees. This indicates that the specimens had an orthorhombic or pseudo-orthorhombic unit cell with a 10_3_ helical conformation, indicating an α-form crystal. The peak intensities of the sc-PLA for both XRD and FTIR results were amplified by increasing the number of printing steps with no observation of any peak indicating homopolymers. Furthermore, the thicknesses of the PLA composites increased with the increasing number of steps (220, 600, and 980 nm for 2, 5, and 10 steps, respectively). As shown in Figure 9b, the XRD patterns analyzed the effect of the number of printing cycles on the structural sc-PLA fabricated by inkjet printing. The L-D LbL-1 specimen, which was fabricated with one cycle of printing, showed two peaks at 12° and 17°, indicating stereocomplex crystals and α-form crystals, respectively. In contrast, homocrystallite formation was no longer observed with increasing cycle numbers for the printed specimens. The intensity of the peaks corresponding to β-form stereocomplex crystals was higher at 2θ = 12°, 21°, and 24° with an increasing cycle number, instead of the appearance of a peak at 2θ = 17°. Consequently, the higher cycle of inkjet printing could render sc-PLA of higher purity and increase its crystallinity. The right graph in Figure 9b shows the XRD patterns of the PLA specimens prepared using PLLA/PDLA mixed solutions. All specimens prepared by inkjet printing using a PLLA and PDLA mixed solution showed no significant increase in the peak intensity despite increasing the number of printing cycles. Furthermore, it only exhibited peaks with no homocrystallites at 12°, 21°, and 24°.

Akashi et al. (2014) reported an advanced method that models drugs such as an 8-mer peptide, ovalbumin (OVA), where proteins could be loaded onto the sc-PLA substrate based on inkjet printing technology [45]. The sc-PLA composites could encapsulate drugs through alternate overprinting of PLLA, PDLA, and drugs on one substrate (Figure 10a). PLLA and PDLA were dissolved in chloroform at a concentration of 0.5 mg/mL. Each polymer solution was alternately printed onto the same substrate. First, the PLLA solution was printed onto a glass substrate and dried at room temperature, and then the PDLA solution was printed onto the substrate. Finally, 0.1 mg/mL of the drug, like OVA-protein, OVA-NPs, or peptide was printed onto the printed sc-PLA composite. A single cycle for the entire inkjet-printing process comprised these three steps. This process was conducted for up to a maximum of 10 cycles. As shown in Figure 10b, 25% of the peptide in the sc-PLA carrier was released in the initial 1 h, and 50% of the peptide was released after 1 day. Then, 90% of the peptide was released for 5 days via diffusion. In contrast, 40% of ovalbumin in the sc-PLA carrier was released for 1 day, and the remainder of the loaded protein failed to be released. This was assumed to be due to the aggregation of ovalbumin proteins in the sc-PLA matrix. In contrast, the sc-PLA with OVA-NPs group, which was printed by ovalbumin encapsulated nanoparticles (OVA-NPs), released 30% of the protein for the initial 1 day, and subsequently the remainder of the contents could be sustainably released during the following 30 days. It was considered that denaturation and aggregation of protein could occur to a lesser extent in the sc-PLA composite, because encapsulation of ovalbumin in the nanoparticle could offer resistance to protein denaturation and stability to the carrier. As shown in Figure 10c, the in vitro release profile of sc-PLA with ovalbumin encapsulated nanoparticles and homopolymer (PLLA and PDLA) with ovalbumin encapsulated nanoparticles was analyzed. All groups exhibited burst release kinetics of the protein for 24 h, and their release rates gradually decreased. The sc-PLA composite group showed a lower cumulative release of ovalbumin than the homopolymers after 24 h (28 vs. 35 vs. 41% for sc-PLA-OVA-NPs, PLLA-OVA-NPs, and PDLA-OVA-NPs, respectively). Furthermore, the sc-PLA-OVA-NPs had a lower release rate with sustainable release compared to the other groups. It was supposed that sc-PLA had superior hydrolysis resistance and carrier stability owing to its higher crystallinity compared to homopolymers.

Recently, Akashi et al. (2017) reported that sc-PLA composites could be conjugated to benzyl alcohol and 3,4-diacetoxycinnamic acid (DACA) of bio-based aromatic compounds at the hydroxyl groups of both terminals using inkjet printing [46]. This DACA conjugation enhanced the thermal stability of sc-PLA by increasing the thermal decomposition temperature by 10% to above 90 °C. Consequently, inkjet printing can be considered as an innovative technology for fabricating sc-PLA scaffolds rapidly and easily based on LbL deposition and assembly. Furthermore, it has the advantage of versatile control of the shape, thickness, and amount of printed scaffolds composed of sc-PLA. Therefore, this technology has the potential to bring innovation to some fields, such as tissue engineering and biomaterials, if it can be converged with 3D printing for freely customized fabrication of scaffolds and substrates.

### 2.5. Stereocomplex Hydrogel

A hydrogel is a network of cross-linked polymer chains. Hydrogels have been frequently used as scaffolds in tissue engineering and drug carriers and are known as the first biomaterials to be used in the human body. S.J. de Jong et al. reported that a stereocomplex hydrogel could be synthesized by mixing dextran-grafted L-lactate and D-lactate in an aqueous solution [47]. The stereocomplex hydrogel could encapsulate the IgG and lysozyme of the model protein, and the loaded contents were released by Fickian diffusion for six days. Even after the release of sensitive proteins from the gel, the stereocomplex hydrogel played the role of a stable protein carrier as well as the maintenance of enzymatic activity. Subsequently, Hennink et al. fabricated a stereocomplex hydrogel by mixing dextran-L or D-lactate without organic solvents or crosslinking agents in an aqueous environment, as shown in Figure 11a [48]. Enantiomeric PLA oligomers grafted to dextran did not require artificial agents, because they were already crosslinked by stereocomplexation. Moreover, this stereocomplex hydrogel has the advantage of full biodegradability and clinical safety, because dextran is a non-toxic water-soluble polymer. Figure 11b shows the rheological properties of dex-(L)lactate and a mixture of dex-(L)lactate and dex-(D)lactate. A mixture of enantiomers exhibited a growth in the storage modulus (G’) and a reduction of tan δ with time, whereas dex-(L)lactate showed no change in G’ and tan δ with time. These results indicate that the hydrogel network was formed, and that the polymer had a more elastic property. This was presumed to be due to self-assembly between the chains of L-lactate and D-lactate via stereocomplexation. As shown in Figure 11c, the lysozyme was released from the dex-lactate hydrogel faster than IgG in the same carrier during the initial stage. The hydrogel with higher polydispersity (PDI) of the lactate graft showed faster release of the two model proteins compared to those with lower PDI. The results showed that all groups exhibited complete release of the loaded contents from the hydrogel after eight days and retained enzymatic activity. Hence, the sc-PLA hydrogel could encapsulate proteins and control release. It is potentially applicable to drug carriers with good biocompatibility and gelation behavior.

A strategy using in situ gelling systems has been used for the transformation of drug/polymer precursor complexes from solution after injection into the human body to gel form by physiological conditions of target tissues or artificial stimuli, such as pH or temperature change, UV irradiation, solvent exchange, catalytic ions, or molecules [49,50,51]. Generally, an in situ gelling hydrogel can be synthesized by various chemical reactions, including enzyme-catalyzed cross-linking, Schiff-base reaction, photo-induced polymerization, and Michael-type addition [52,53,54,55]. This drug delivery system can prevent adverse events in non-target tissues with improved availability of administration. Based on this strategy, sc-PLA-based hydrogels have the potential to improve the mechanical strength and durability and delay the degradation rate of the carrier induced by stereocomplexation in the future.

## 3. Biomedical Applications

### 3.1. Anti-Cancer Therapy

Recently, numerous studies have reported that biodegradable polymeric nanocarriers possess a superior stealth function for the detection of the reticuloendothelial system in the human body and exhibit an enhanced permeability and retention effect [56,57,58,59,60,61]. In particular, these polymeric nanoparticles are capable of having several beneficial properties, such as biomimetics, stimuli-sensitivity, easy modification, and exquisite target specificity compared to carriers composed of other materials. These advantages support the possibility that stereocomplex nanoparticles can be used as a promising approach in the field of anti-cancer therapy.

Goldberg proposed that oligomers of PDLA were used for the formation of stereocomplex crystals with L-lactate in the human body to induce lactate deficiency in cancer cells [62]. This study suggested that tumor growth could be inhibited and terminated by stereocomplexation through lactate deficiency to retain the electrical neutrality of tumors. In preliminary experiments, it was demonstrated that high concentrations of PDLA could induce stereocomplexation with lactate in the body, which exhibited cytotoxic effects on the tumor. Li et al. (2016) fabricated sc-PLA-coated nanoparticles with multiple functionalities for a highly tunable drug delivery system via simple LbL self-assembly [14]. TEM images show that the nanoparticles had a spherical shape with a core–shell structure and a diameter of approximately 190 nm (Figure 12b). The in vitro drug release profiles of doxorubicin (DOX)-loaded nanoparticles were analyzed at different pH and temperature conditions to demonstrate the ability to adjust the drug delivery of the sc-PLA-coated nanoparticles. As shown in Figure 12c, the drug release rate was significantly reduced with increasing pH from 3.5 to 7.4. The cumulative release of DOX was 73.1 vs. 55.6% at pH 3.5 and 7.4 after 12 h, respectively. This is because acidic conditions could stimulate protonation of the tertiary amine groups in the outermost layer of the sc-PLA-coated nanoparticles, resulting in the swelling of nanoparticles. Furthermore, a lower pH yields the nanoparticles pH-responsive properties through an increase in the solubility of DOX, which retains phenols and amines. In addition to pH, the cumulative release of DOX was 39.8 vs. 60% at 37 and 20 °C after 12 h, respectively. A lower temperature condition could facilitate a more rapid release via swelling of the outer layer of sc-PLA-coated nanoparticles. Consequently, the nanoparticles based on sc-PLA could tune the rate and amount of drug release, depending on the physiological conditions. To identify the cytotoxicity of DOX-loaded sc-PLA-coated nanoparticles to breast cancer cells, MCF-7 cells were incubated with free DOX and two types of DOX-loaded sc-PLA nanoparticles (Figure 12d). During the incubation of cancer cells with free DOX, most of the DOXs were localized in the cell nuclei, instead of the cytoplasm. In contrast, the DOXs released from all sc-PLA particles infiltrated the cells, and their accumulation was significantly increased. These results indicated that sc-PLA-coated nanoparticles could effectively act as anti-cancer drug carriers by improving the cell uptake efficiency.

Brzeziński et al. synthesized DOX-loaded stereocomplexed microspheres using spontaneous precipitation after the polymerization of L-proline-functionalized PLLA and PDLA via coordination polymerization (Figure 13a) [63]. Therein, the size of the microparticles was dependent on whether the L-proline end groups were blocked or unblocked in the microspheres. Based on this correlation, they obtained spherical microspheres with various sizes ranging from 0.5 to 10 μm through adjusting the solvent and functionalization. In the in vitro release profiles of DOX, stereocomplexed microspheres with Boc-protected L-proline exhibited cumulative release within 10% of loaded contents for 100 h, while microspheres with unblocked L-proline showed a faster cumulative release of 41–81% of loaded contents for the same duration (Figure 13b). This can be attributed to the low surface area of the Boc-protected group, which delayed hydrolysis. In contrast, the unblocked group could easily release DOX due to the localization of DOX on the surface of the microsphere, resulting in an initial burst release. To evaluate the cytotoxicity of the DOX-loaded stereocomplexed microspheres, A549 lung cancer cells were incubated with medium extracts of the two types of microspheres. Figure 13b shows a very slow reduction in the cell viability of cancer cells cultured with Boc-protected microspheres after 24 h, whereas the cell viability of the cancer cells cultured with unblocked microspheres was dramatically decreased after incubation for 2 h and decreased to below 25% after 24 h. This can be attributed to the difference in drug release rate between the blocked and unblocked microspheres. In particular, the microspheres prepared in tetrahydrofuran (THF) showed a significantly high anti-proliferative effect.

In addition, Brzeziński et al. successfully fabricated stereocomplexed micelles with β-cyclodextrin (β-CD) core as an intracellular drug carrier [64]. Hence, micelles stability can be improved, and in vitro release rate of DOX from supramolecular nanocarriers can be controlled. The stereocomplexed micelles with DOX efficiently inhibited the proliferation of HeLa (cervical cancer) and K562 (chronic myelogenous leukemia). This study suggested that the supramolecular interactions facilitate effective establishment of drug delivery system as an anti-cancer therapy.

### 3.2. Tissue Engineering

Tissue engineering is an increasingly popular next-generation biomedical technology to treat defects and malfunctions in human organs. This technology has the potential to expand medical coverage and resolve problems, such as a lack of organ donation and transplant rejection [65,66,67,68,69]. Scaffolds that are suitable for tissue engineering require high biocompatibility, biodegradability, good processability, mechanical properties similar to those of native tissue, and proper flexibility [70,71,72,73,74,75,76,77]. sc-PLA has been extensively applied in tissue engineering owing to its excellent biocompatibility, full biodegradability, improved mechanical properties, and thermal stability. In particular, it is suitable for scaffolds that require robust properties for bone, cartilage, and orthopedic implants. V. Katiyar et al. (2017) fabricated orthopedic implants based on nano-hydroxyapatite (n-HAP)-grafted sc-PLA composites using 3D printing [78]. As shown in Figure 14a, n-HAP-grafted PDLA was polymerized by in situ ring-opening polymerization, and the sc-PLA/n-HAP filament with a diameter of approximately 1.6 mm for 3D printing was fabricated by melt mixing with PLLA in a twin-screw extruder. A middle phalanx bone composed of filaments was successfully manufactured using 3D printing. As shown in FE-SEM images, the fractured Sc-PLA/n-HAP nanocomposites exhibited a smooth surface and uniform dispersion of n-HAP of 60 nm size (Figure 14b). The n-HAP provided a reinforcement effect and expansion of the surface area as a filler in the sc-PLA matrix. As shown in Figure 14c, sc-PLA/n-HAP of 2.5% increased the ultimate tensile strength up to a maximum of 16% above that of neat sc-PLA (40.2 vs. 33.8%, respectively). This is because an increase in intermolecular bonding and cross-linking in stereocomplex crystals, together with strong interfacial bonding between the polymer matrix and the filler, could increase its crystallinities. Furthermore, the ductility of sc-PLA increased by the addition of n-HAP fillers, which increased the elongation at the break of sc-PLA/n-HAP up to a maximum of 131.6 %. Improving ductility could enhance the durability of biocomposites resulting from the prevention of fracture and abruption, thus expanding its application for implants that require robust resistance for high loads.

Subsequently, V. Katiyar et al. (2019) focused on the synthesis of linear block copolymers composed of hard and soft segments of PLLA/PDLA and PCL [79]. Diblock and stereotriblock copolymers were successfully polymerized with PCL as a macroinitiator using sequential ring-opening polymerization, as shown in Figure 15a. Values of the tensile strength and elongation at the break for diblock copolymers were improved from 14.8 to 28.9 MPa and from 6.4 to 17.8%, respectively, with an increase of the block length of PDLA (Figure 15b). The synthesized materials were thermally processed based on injection molding to manufacture cancellous and cortical bone screws, which are considered as orthopedic fixation devices, as shown in Figure 16a. In a study on thermo-mechanical stability, cancellous bone screws consisting of sc-PLA/PCL blends could stably maintain their shape and structure at 121 °C for 60 min, more than those of commercial homo PLA (Figure 16b). This is because the stereocomplex crystallites of the hard segment in the blend copolymer improved the thermal resistance of the scaffold. Consequently, the sc-PLA and PCL in the sc-PLA block copolymer enabled the scaffold to increase mechanical and thermal stability, and to reduce brittleness of PLA by its plasticization effect; thus, scaffolds composed of these biomaterials are considered suitable for biomedical implants with good clinical outcomes.

Enhancing mechanical properties of biodegradable polymers is critical to biomedical fields, such as bone fixation. Numerous processing methods have been developed to improve the strength of the polymers. Many studies have demonstrated that solid-state drawing (SSD) can induce self-reinforcement through the maximization of macromolecular chain orientation in polymeric materials [80,81,82]. Im et al. (2016, 2017) determined that the tensile strengths of PLLA monofilaments and films could be increased up to two- and nine-fold, respectively, by increasing the draw ratio using a directly designed processing machine for the SSD method (Figure 17a) [83,84]. Furthermore, this study showed that solid-state drawn PLLA enhanced blood compatibility and cell adhesion. Recently, Li et al. (2021) successfully oriented shish-kebab crystals in sc-PLA using the SSD method, as shown in Figure 17b [85]. The oriented sc-PLA scaffold had a tensile strength of 373 MPa and an elongation of 9%. This processing could lead to fibrous crystals of shish and kebabs with parallel lamellar microstructures along the direction of the drawing. This shish-kebab microstructure with a specific topography could provide a self-reinforcing effect and prevent cracking and collapse of aligned kebabs into biomaterials based on sc-PLA.

C. Wang et al. (2019) synthesized injectable thermogels based on the sterocomplex 4-arm poly(ethylene glycol)-polylactide (PEG-PLA) and cholesterol-modified 4-arm PEG-PLA for optimized cartilage regeneration (Figure 18) [86]. The cholesterol-modified sc-PLA gels exhibited improved mechanical strength, lower critical gelation temperature, higher chondrocyte proliferation, and slower degradation than unmodified specimens. Moreover, cholesterol-modified sc-PLA gel-loaded chondrocytes showed considerably more cartilage-like tissues than fibrous- and bone-like tissues. This is attributed to the improved mechanical properties and microstructure induced by cholesterol modification of the sc-PLA gel.

### 3.3. Anti-Microbial Effect

Biomaterials and surgical implants based on the sc-PLA material are increasingly being applied for the above-mentioned biomedical applications, including tumor treatment and tissue engineering. However, preventing contamination from foreign microorganisms, such as bacteria and viruses is essential for the application of sc-PLA as a biomaterial in the human body [87,88,89]. Sterilization is necessary to prevent contamination before medical surgery, albeit it has been shown that contamination from bacteria such as *Staphylococcus aureus* (*S. aureus*) most frequently occurs during surgery [90,91,92]. Therefore, biomaterials based on sc-PLA are critical for securing anti-microbial effects to prevent bacterial proliferation to decrease adverse events and maximize clinical efficacy. Normally, biomaterials are sterilized by ethylene oxide gas, gamma ray irradiation, dry-heat sterilizer, microwave, autoclave, and supercritical carbon dioxide (CO_2_) [93,94,95,96,97]. Unfortunately, these methods have remaining concerns not only regarding changes in the inherent properties of the material during the sterilization process, but also the difficulty in preventing secondary contamination. Thus, biomaterials must possess sustained anti-microbial effects to inhibit external microorganisms before and after implantation in the body. Spasova et al. (2010) fabricated electrospun sc-PLA fibers with antibacterial and hemostatic effects using diblock copolymers composed of poly(N,N-dimethylamino-2-ehtylmethacrylate) (PDMAEMA) [98]. After the incubation of mats composed of these fibers with *S. aureus* and *Escherichia coli*, the adhesion of these bacteria was observed. Consequently, sc-PLA mats containing PDMAEMA significantly inhibited bacterial adhesion and proliferation on the surface and exhibited effective antibacterial effects, while the control group showed bacterial adhesion and biofilm formation on the surface. This was attributed to the surface of tertiary amino groups from PDMAEMA blocks [99,100,101]. This surface modification strategy for antibacterial effects could remove concerns about contamination in surgical procedures and sustain its efficacy in the human body. Ajiro et al. (2016) polymerized PLLA and PDLA using catechin (CT) as an initiator precursor, which is an antibacterial compound [102]. Figure 19a shows the polymerization of CT-conjugated PLLA and PDLA at the chain end groups. Lactide was polymerized with benzyl catechin (BnCT), and then CT was chemically combined with PLAs to protect the phenolic hydroxyl groups. To assess antibacterial properties of the polymerized products, the ratio of total viable bacteria was calculated after 24 h according to the JIS Z 2801 test, as shown in Figure 19b. The CT-absorbed substrate reduced the ratio of total viable counts by 20% compared to the control. In contrast, both PLLA-CT and PLA-CT-SC substrates had approximately 50% significantly lower values than those of the control group. As shown in Figure 19c, the counts of killed bacteria per CT unit were compared with those of the CT-absorbed substrate, PLLA-CT, and PLA-CT-SC. Both PLLA-CT and PLA-CT-SC groups showed antibacterial properties mainly induced by the phenolic hydroxyl groups of CT. The antibacterial effect was dependent on the amount of CT, and it could be maintained for long-term use.

Y. Li et al. (2013) suggested a novel approach for synthesizing charged hydrogels based on non-covalent interactions for disrupting biofilms and microbes [103]. They fabricated the antimicrobial gel by stereocomplexation between PLLA-PEG-PLLA and a charged PDLA-polycarbonate-PDLA (PDLA-CPC-PDLA) triblock polymer in aqueous solution. At physiological temperature (37 °C), the physical properties of the stereocomplex hydrogel were transformed into shear thinning behavior with supramolecular fiber and ribbon-like structures. This improved antimicrobial activity of the cationic hydrogel against diverse pathogenic microorganisms, such as fungi and both Gram-positive and Gram-negative bacteria. Up to 60% film biomass of *S. aureus*, *E. coli*, *Candida albicans* (*C. albicans*), and Methicillin-resistant *S. aureus* (MRSA) were eliminated after hydrogel treatment. In the safety test for skin sensitization, acute dermal toxicity, and skin irritation, the stereocomplex hydrogels appeared to provoke no adverse events in animal models of rats, guinea pigs, and rabbits. L. Mei et al. (2018) fabricated hybrid nanofibers by electrospinning in addition to PLA stereoisomers for inducing stereocomplexation between the stereoisomer chains and addition of chlorogenic acid (CA) of an antibacterial agent [104]. To prevent damage of the agent, the stereocomplex nanofibers were electrospun at relatively lower temperature (65 °C). The antibacterial fibers could effectively remove both gram-positive and gram-negative bacteria by quickly released CA within a few hours. The fibers based on sc-PLA have a potential to be applied in various fields such as filter, masks, and packages, owing to enhanced mechanical and thermal properties as well as full biodegradability. Recently, Y. Ren et al. (2019) reported successful fabrication of a novel eco-friendly sc-PLA nanofiber by electrospinning, with both functions of adsorption of heavy metal ions and inhibition of bacterial growth [105]. In the fabrication process, an antibacterial agent called HTA, which could be synthesized from tannic acid and hexamethylenediamine, was loaded into the products to provide an antibacterial effect. Furthermore, the electrospun nanofibers had improved tensile strength and Young’s modulus, as well as thermal resistance owing to the formation of stereocomplex crystallites. The nanofiber mats based on sc-PLA exhibited excellent abilities for adsorption of Cr(VI) and capture of *E. coli* and *S. aureus*. Cr(VI) was converted to less toxic Cr(III) after its adsorption. This indicates that the heavy metal pollutant can be changed into eco-friendly and stable compounds in nature.

## 4. Conclusions

This review focuses on the research on sc-PLA with biodegradability and superior mechanical and thermal properties. sc-PLA can be used to produce therapeutic carriers in various forms, as it can be processed by micelles, self-assembly, emulsion, and 3D printing. In particular, 3D printing technology has commonly used PLA as filament material; however, sc-PLA has not been used in industrial practice to date. Because of these application fields and the extensive potential of 3D printing, inkjet printing using sc-PLA with improved characteristics must be a versatile technology for simultaneously inducing both stereocomplexation and fabrication in the future, if inkjet printing can be suitably converged with 3D printing techniques. In biomedical applications, sc-PLA nanoparticles are potentially promising carriers for anti-cancer therapy and scaffolds for tissue engineering, as numerous studies have demonstrated that they could be improved by diverse methods, including surface modification and co-polymerization, and attain additional functions, such as anti-microbial effects and immune stealth. sc-PLA nanoparticles effectively encapsulate therapeutic anti-tumor agents, such as DOX, and specifically transfer the agents to target lesions. This review suggests the fields for potential sc-PLA material applications by presenting current research trends. sc-PLA, as an advanced material from commonly used PLA, is expected to become a next-generation polymeric material owing to its excellent biocompatibility, biodegradability, and mechanical and thermal properties. 

## Figures and Tables

**Figure 1 molecules-26-02846-f001:**
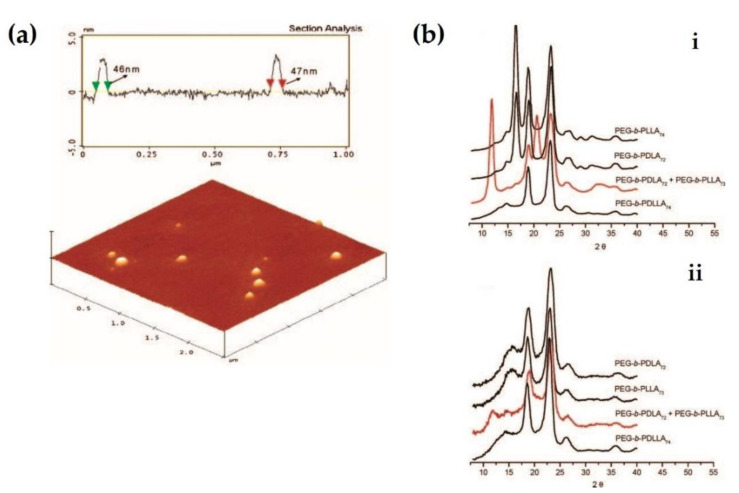
(**a**) AFM image of stereocomplexed micelle induced by mixing of PEG-*b*-PDLA_72_ and PEG-*b*-PLLA_73_ at equal proportions. (**b**) XRD patterns of PEG-*b*-PLA films (**i**) and stereocomplexed micelles composed of equimolar amounts of PEG-*b*-PDLA_72_ and PEG-*b*-PLLA_73_ (**ii**) [21], Copyright 2005. Reproduced with permission from the American Chemical Society.

**Figure 2 molecules-26-02846-f002:**
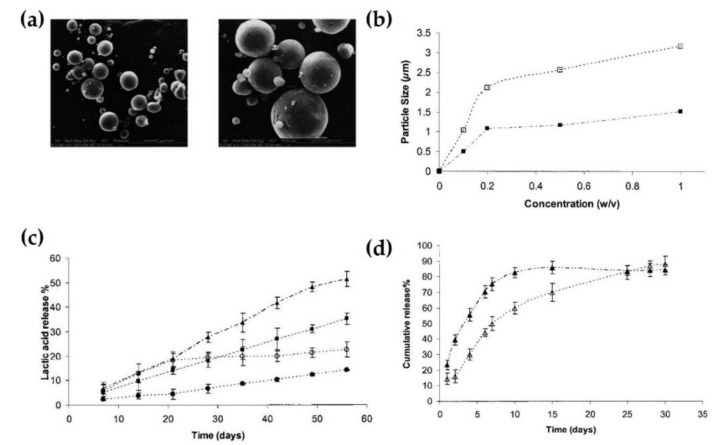
(**a**) Scanning electron microscopy (SEM) images of stereocomplex nanoparticles composed of PLLAx-PEG2000-PLLAy with D-PLA (x + y = 25, left) and PLLAx-PEG2000-PLLAy with 10% *w*/*w* dexamethasone phosphate with D-PLA, respectively (right). (**b**) Stereocomplex particle size at different concentrations. Closed and open squares depict PLLAx-PEG2000-PLLAy with D-PLA (x + y = 25) and PLLAx-PEG2000-PLLAy with D-PLA (x + y = 50) specimens, respectively. (**c**) HPLC analysis for lactic acid release from copolymers and stereocomplexes. Open and closed circles depict PLLAx-PEG2000-PLLAy (x + y = 25) and PLLAx-PEG2000-PLLAy (x + y = 50), respectively. Closed squares and triangles depict PLLAx-PEG2000-PLLAy (x + y = 25) with D-PLA specimen and same material containing 10% *w*/*w* dexamethasone phosphate, respectively. (**d**) HPLA analysis for in vitro release of dexamethasone from stereocomplexes. Closed and open triangles depict specimens composed of PLLAx-PEG2000-PLLAy (x + y = 25) with D-PLA and PLLAx-PEG2000-PLLAy (x + y = 50) with D-PLA, respectively. This experiment was conducted in phosphate buffer (pH 7.4) at 37 °C [26], Copyright 2005. Reproduced with permission from WILEY-VCH Verlag GmbH & Co.

**Figure 3 molecules-26-02846-f003:**
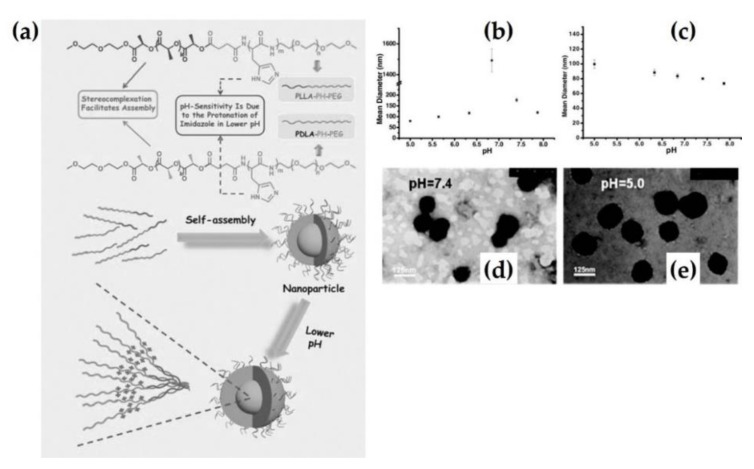
(**a**) Formation process schematic of mPEG_45_-PH_30_-PLLA_82_/mPEG_45_-PH_30_-PDLA_82_ stereocomplex nanoparticles. Mean diameters of mPEG_45_-PH_30_-PLLA_82_ (**b**) and mPEG_45_-PH_30_-PLLA_82_/mPEG_45_-PH_30_-PDLA_82_ stereocomplex nanoparticles (**c**) with various pH conditions. TEM images of the stereocomplex nanoparticles at pH 7.4 (**d**) and pH 5.0 (**e**) [27], Copyright 2012. Reproduced with permission from WILEY-VCH Verlag GmbH & Co.

**Figure 4 molecules-26-02846-f004:**
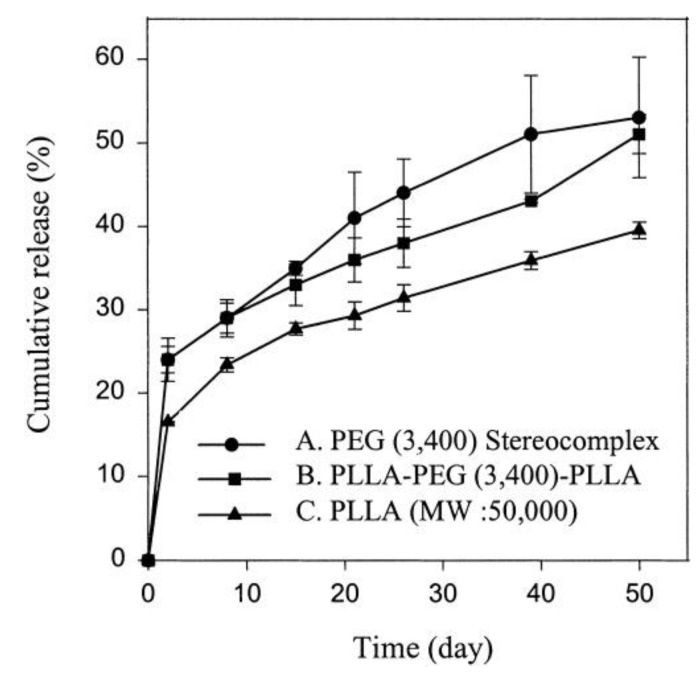
In vitro BSA release profile from three specimens [29], Copyright 2000. Reproduced with permission from John Wiley & Sons, Inc.

**Figure 5 molecules-26-02846-f005:**
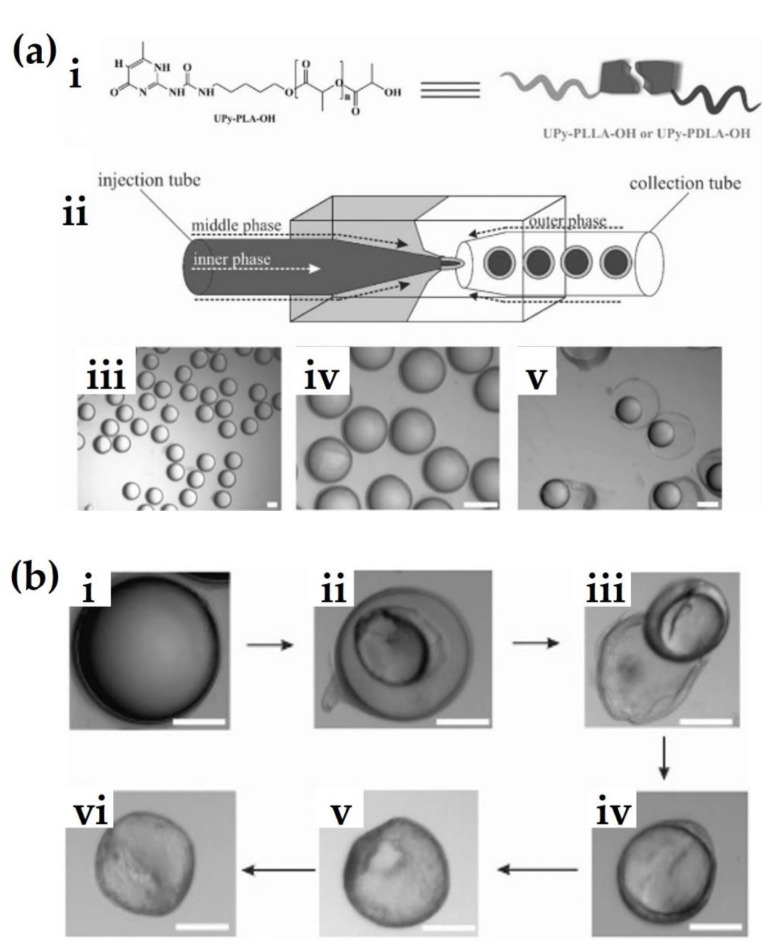
(**a**) Schematic illustrations of UPy-PLA-OH chemical structure (**i**) and microfluidic device for induction of W/O/W emulsion droplets (**ii**). Optical microscope observations of monodisperse stereocomplexed drops (**iii**,**iv**), and UPy-PLA-OH unstable drops (**v**). Scale bars: 200 μm. (**b**) Change of structural arrangement of W/O/W double emulsion droplets. (**i**) sc-PLA-UPy double emulsion droplet, (**ii**,**iii**) separation of microcapsule in water phase, (**iv**–**vi**) shell solidification of the microcapsule induced by evaporation. Scale bars: 100 μm [38], Copyright 2017. Reproduced with permission from WILEY-VCH Verlag GmbH & Co.

**Figure 6 molecules-26-02846-f006:**
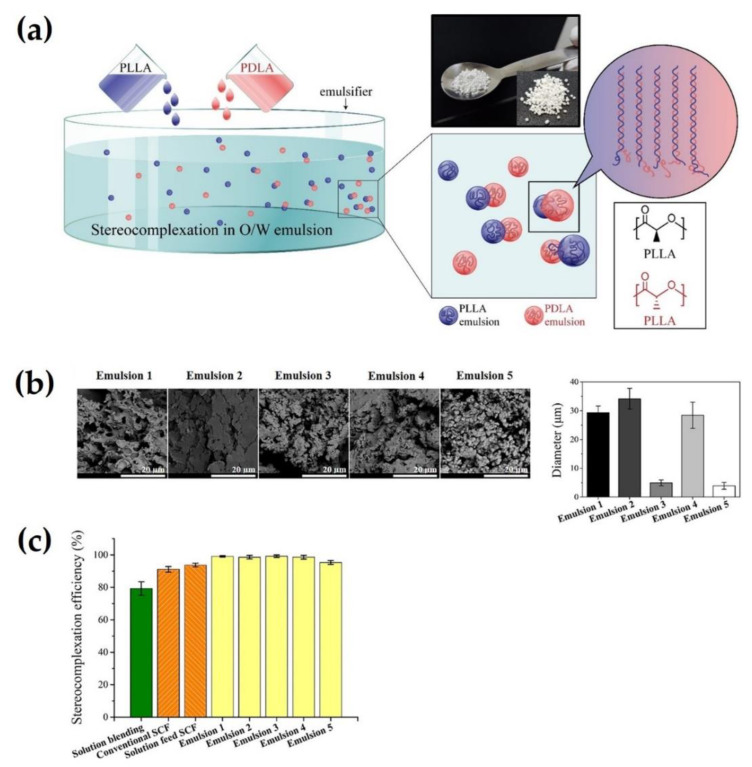
(**a**) Scheme for O/W emulsion blending approach for inducing sc-PLA. Inserted image exhibits sc-PLA particles prepared by emulsion blending. (**b**) SEM observations (X10000, left) and diameters (right) of sc-PLA particles prepared by various O/W emulsion blending named Emulsion 1-5. (**c**) Comparison of stereocomplexation efficiency (%) of sc-PLA particles prepared Emulsion 1-5. SCF: supercritical fluid technology [43], Copyright 2020. Reproduced with permission from the American Chemical Society.

**Figure 7 molecules-26-02846-f007:**
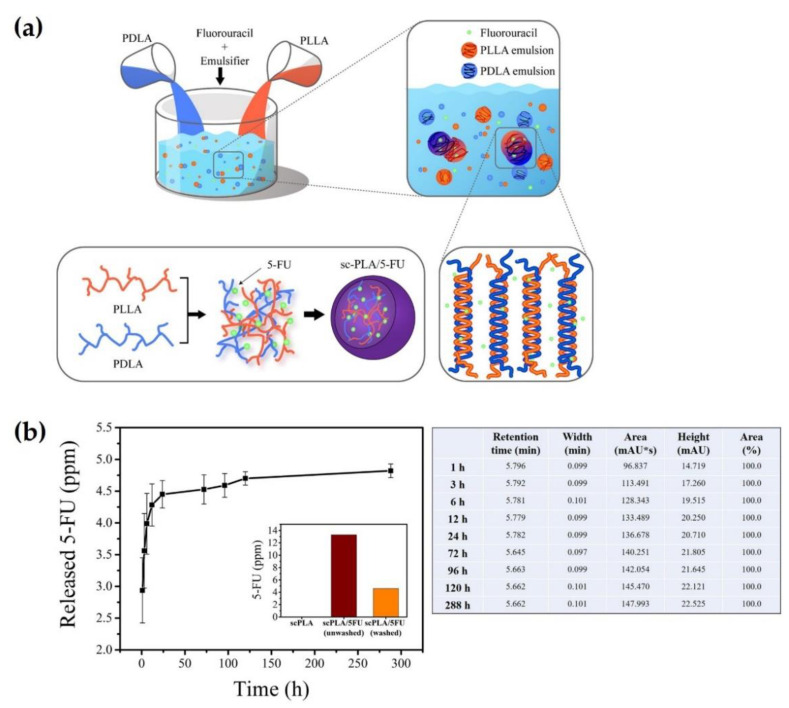
(**a**) Schematic illustration of O/W emulsion blending for simultaneously inducing stereocomplexation and infiltration of 5-FU drugs. (**b**) Release profile of 5-FU from washed sc-PLA particles prepared by O/W emulsion blending over 12 days. Insert presents graph of released 5-FU concentration from neat, washed, and unwashed sc-PLA/5-FU specimens. Data are plotted as mean values ± standard deviation (SD) (*n* = 3) [43], Copyright 2020. Reproduced with permission from the American Chemical Society.

**Figure 8 molecules-26-02846-f008:**
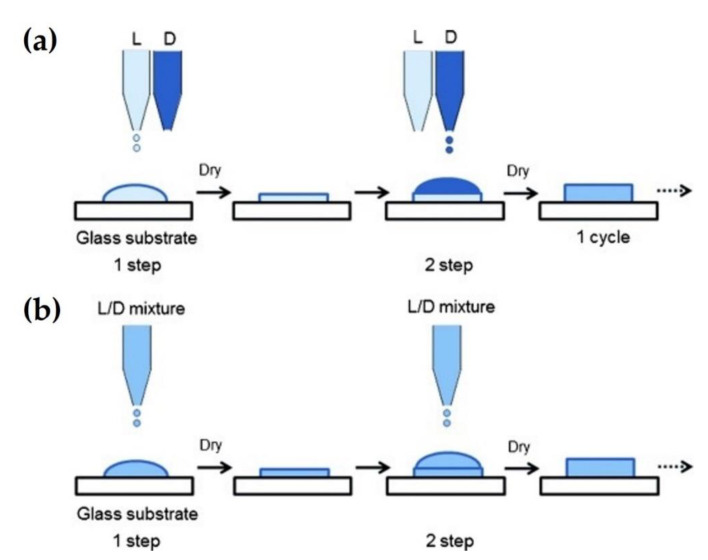
Schematic illustration for sc-PLA formation using the inkjet printing based on LbL deposition. (**a**) First LbL-assembly method; alternately printing of PLLA and PDLA solution onto a substrate in sequence (2 steps = 1 cycle), (**b**) Second LbL-assembly method; simultaneous printing of PLLA/PDLA mixed solution onto a substrate at one time [44], Copyright 2012. Reproduced with permission from WILEY-VCH Verlag GmbH & Co.

**Figure 9 molecules-26-02846-f009:**
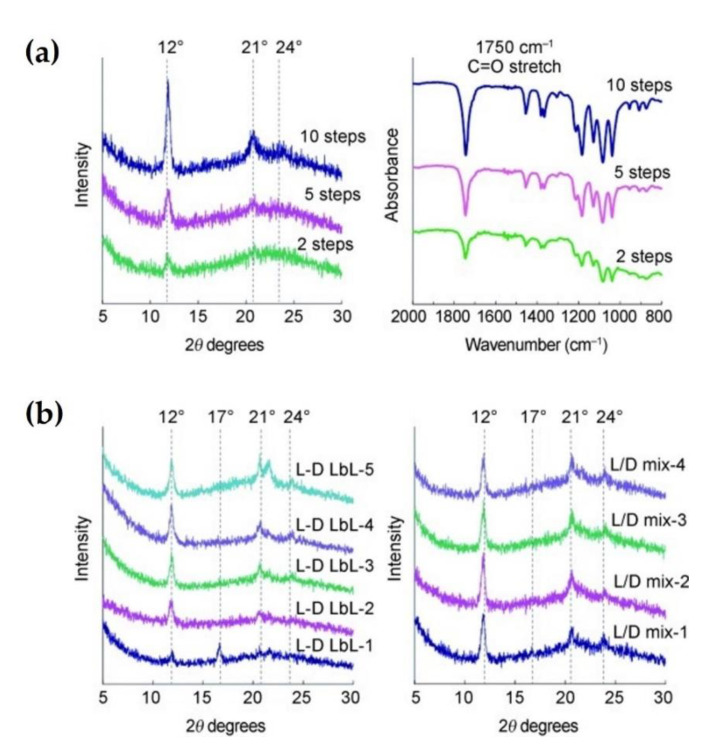
(**a**) XRD patterns (left) and FTIR spectra (right) of sc-PLA fabricated by inkjet printing of the second LbL-assembly method. The steps of three specimens indicate the number of repeated times of the printing. (**b**) Influence of the number of printing cycle on crystal formation of inkjet-printed sc-PLA. XRD patterns of sc-PLA product prepared by first (left) and second (right) LbL-assembly method [44], Copyright 2012. Reproduced with permission from WILEY-VCH Verlag GmbH & Co.

**Figure 10 molecules-26-02846-f010:**
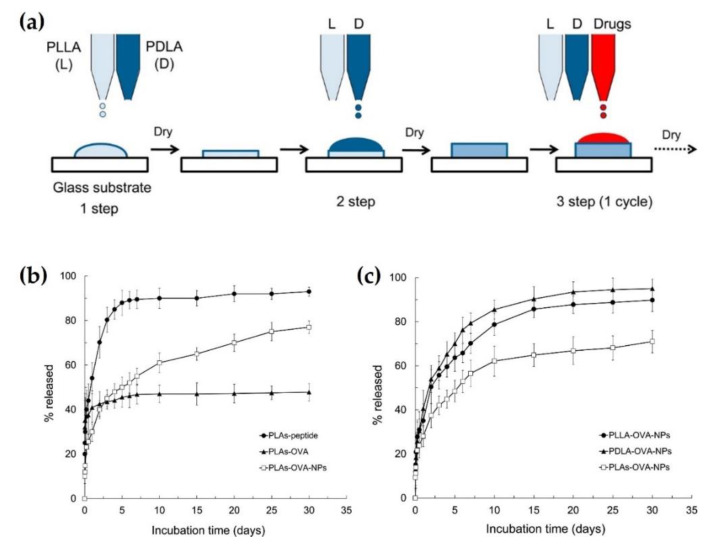
(**a**) Schematic illustration for sc-PLA with drugs composites fabricated by inkjet printing based on LbL-assembly. (**b**) Drug release profiles of the sc-PLA with drug (peptide or OVA) composites in phosphate-buffered saline (PBS). The data were plotted as mean values ± SD (*n* = 3). (**c**) OVA release behaviors from sc-PLA-OVA-NPs (named PLAs-OVA-NPs) and PLLA or PDLA-OVA-NPs in PBS. Results were plotted as mean values ± SD (*n* = 3) [45], Copyright 2014. Reproduced with permission from the American Chemical Society.

**Figure 11 molecules-26-02846-f011:**
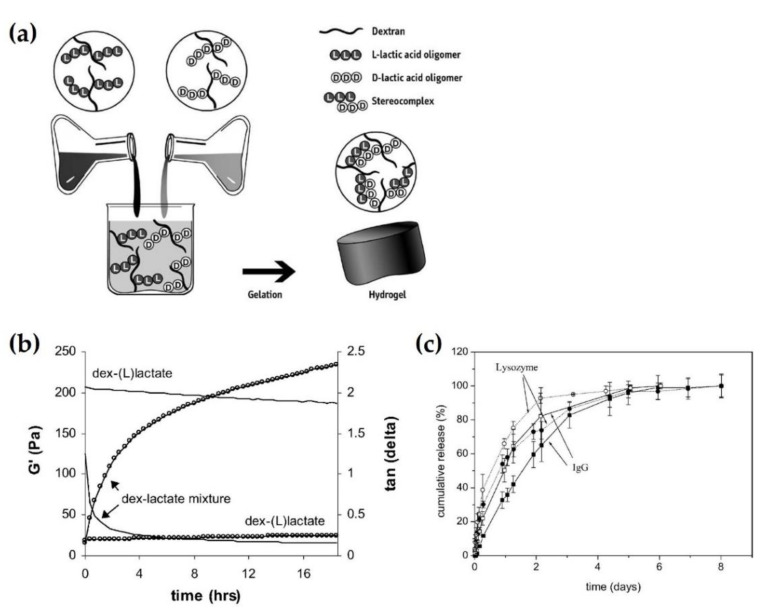
(**a**) Formation of stereocomplex hydrogel schematic via blending dextran-L or D-lactate without organic solvents or crosslinking agents. (**b**) Measurements of rheological properties of dex-(L)lactate and a mixture of dex-(L)lactate and dex-(D)lactate. Open circles and lines in the graph depict G’ and tan δ, respectively [45], Copyright 2003. Reproduced with permission from Elsevier B.V. (**c**) Protein release profiles of lysozyme (indicated by dotted line in the graph) and IgG (indicated by solid line in the graph) from dex-lactate hydrogel exhibiting high (open symbols) and low (filled symbols) PDI. The measurement environment was at pH 7.0 and 37 °C. The data were plotted as mean values ±SD (*n* = 4) [47], Copyright 2001. Reproduced with permission from Elsevier Science B.V.

**Figure 12 molecules-26-02846-f012:**
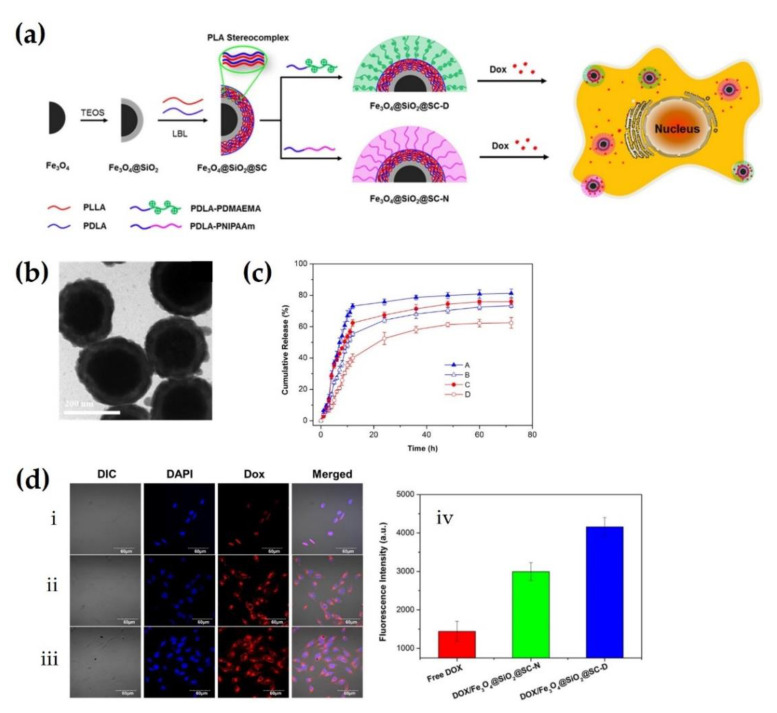
(**a**) Schematic representation for highly tunable sc-PLA coated nanoparticles and drug delivery. (**b**) TEM image of Fe_3_O_4_@SiO_2_@SC-N nanoparticles. Scale bar: 200 nm. (**c**) In vitro drug release profiles of DOX-loaded Fe_3_O_4_@SiO_2_@SC-D nanoparticles in acidic condition of pH 3.5 (A), physiological condition of pH 7.4 (B), and Fe_3_O_4_@SiO_2_@SC-N nanoparticles at 20 °C (C) and 37 °C (D). (**d**) Confocal laser scanning microscope observations of MCF-7 cells incubated with free DOX (**i**), DOX-loaded Fe_3_O_4_@SiO_2_@SC-N (**ii**), and Fe_3_O_4_@SiO_2_@SC-D (**iii**) nanoparticles. Each panel named DIC, DAPI, Dox, and Merged depicts a differential interference contrast (DIC) image, cell nuclei staining by DAPI, DOX fluorescence in cells, and overlay of the all images, respectively. (iv) Comparison of fluorescence intensity in three specimens calculated by ImageJ [14], Copyright 2015. Reproduced with permission from the American Chemical Society.

**Figure 13 molecules-26-02846-f013:**
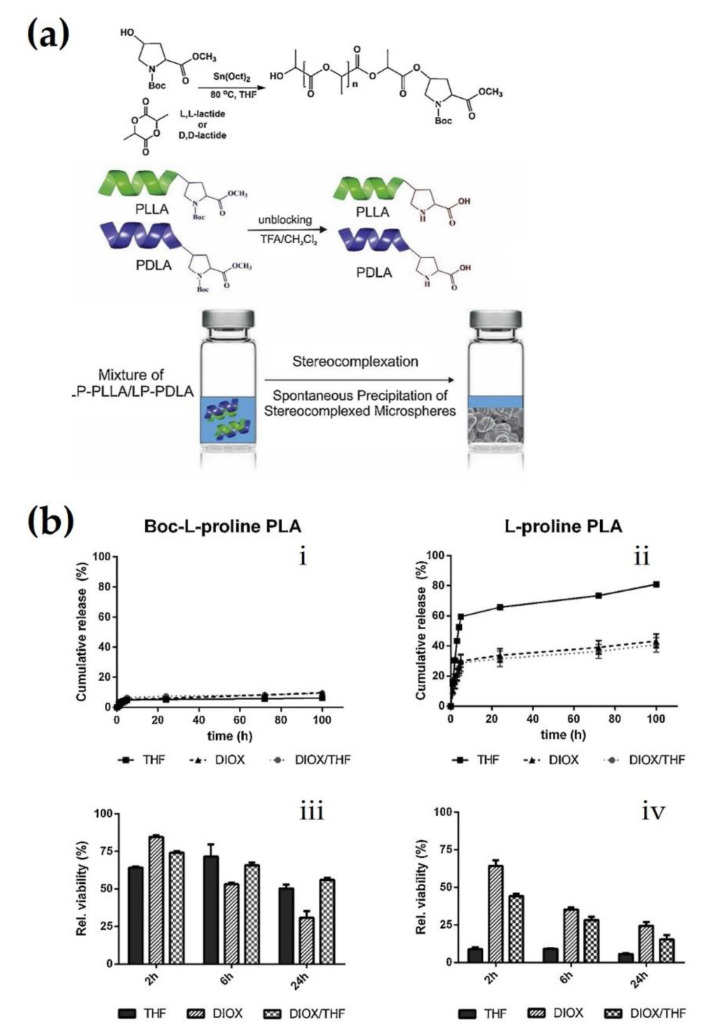
(**a**) Synthesis process of Boc-L-proline functionalized PLLA and PDLA, unblocking of L-proline end groups (top). Schematic illustration for fabrication of sc-PLA microspheres using spontaneous precipitation from mixture of enantiomeric PLAs (bottom). (**b**) In vitro release profiles of DOX from sc-PLA microspheres functionalized with (**i**) Boc-L-proline and (**ii**) L-proline. In vitro cytotoxicity test on A459 cells incubated with medium extracts of the sc-PLA microspheres functionalized with (**iii**) Boc-L-proline and (**iv**) L-proline at incubation times of 2, 6, and 24 h [63], Copyright 2019. Reproduced with permission from Elsevier B.V.

**Figure 14 molecules-26-02846-f014:**
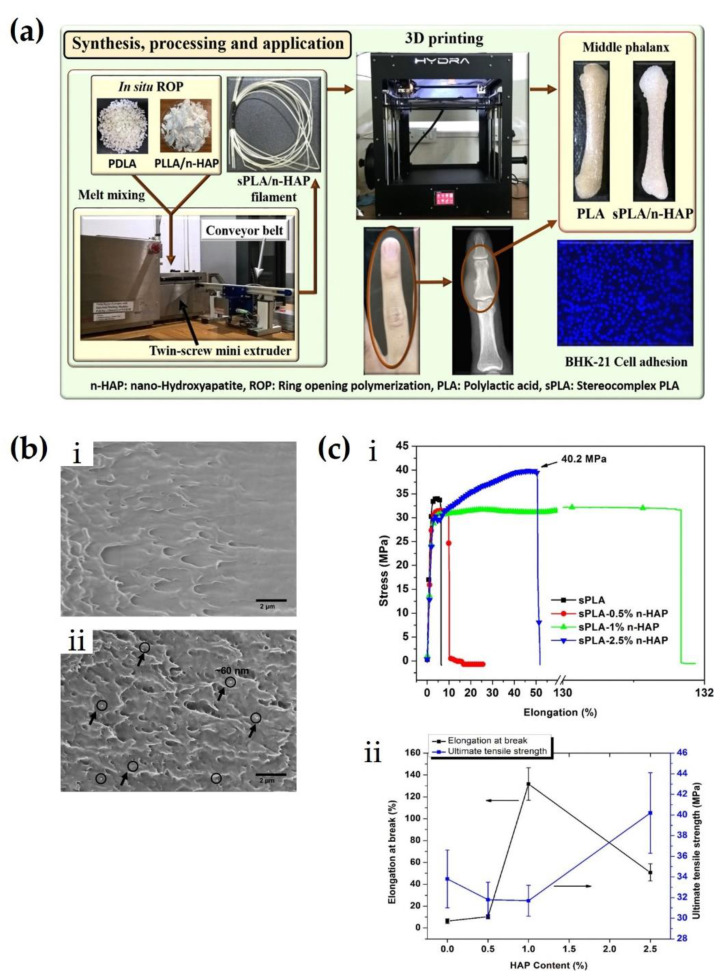
(**a**) A schematic representation for synthesis, processing, and application of sc-PLA/n-HAP. (**b**) Field emission scanning electron microscopy (FESEM) images of fractured surface of (**i**) a neat sc-PLA and (**ii**) a sc-PLA/n-HAP biocomposite. Pointed arrows depict n-HAP particles of approximately 60 nm diameter. (**c**) (**i**) Load–elongation curves of sc-PLA and sc-PLA/n-HAP. (**ii**) Comparison of ultimate tensile strength and elongation at break of sc-PLA and sc-PLA/n-HAP with diverse HAP contents [78], Copyright 2017. Reproduced with permission from the American Chemical Society.

**Figure 15 molecules-26-02846-f015:**
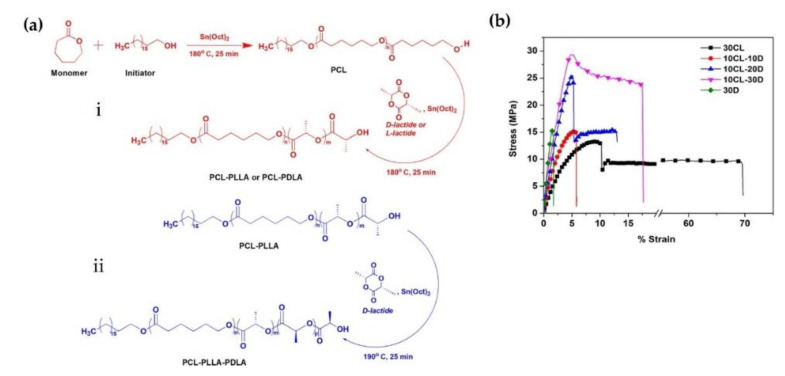
(**a**) Polymerization process of (**i**) diblock and (**ii**) stereotriblock copolymers. (**b**) Stress–strain (SS) curves of homopolymers and diblock copolymers [79], Copyright 2019. Reproduced with permission from the American Chemical Society.

**Figure 16 molecules-26-02846-f016:**
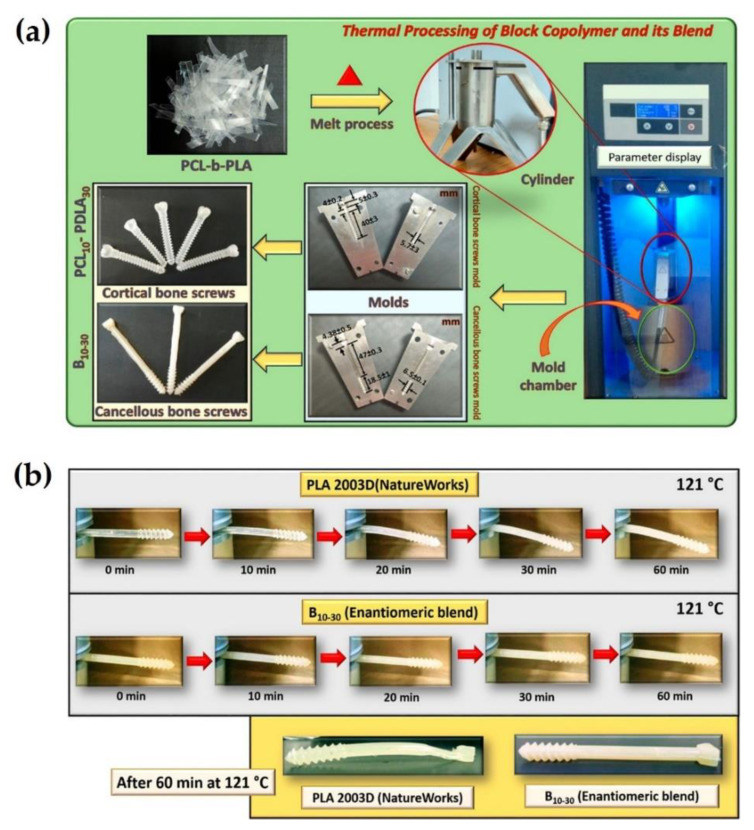
(**a**) Schematic illustration for thermal processing of diblock copolymer and enantiomeric diblock blend for fabrication of cortical and cancellous bone screws as orthopedic fixation devices. (**b**) Comparison of thermo-mechanical stability of cancellous bone screw comprising commercial PLA 2003D (Natureworks) and enantiomeric blend at 121 °C at intervals up to 60 min [79], Copyright 2019. Reproduced with permission from the American Chemical Society.

**Figure 17 molecules-26-02846-f017:**
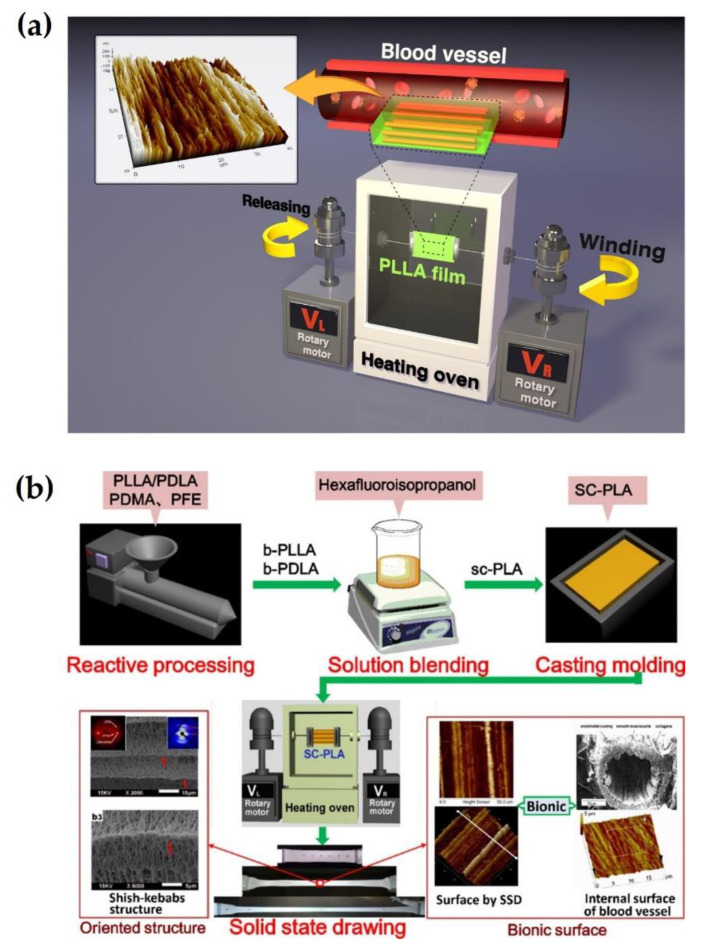
(**a**) Schematic of processing machine for SSD method with PLLA films or filaments [83], Copyright 2016. Reproduced with permission from IOP Publishing, Ltd. (**b**) Schematic representation for preparation process of oriented sc-PLA using SSD method [85], Copyright 2021. Reproduced with permission from the American Chemical Society.

**Figure 18 molecules-26-02846-f018:**
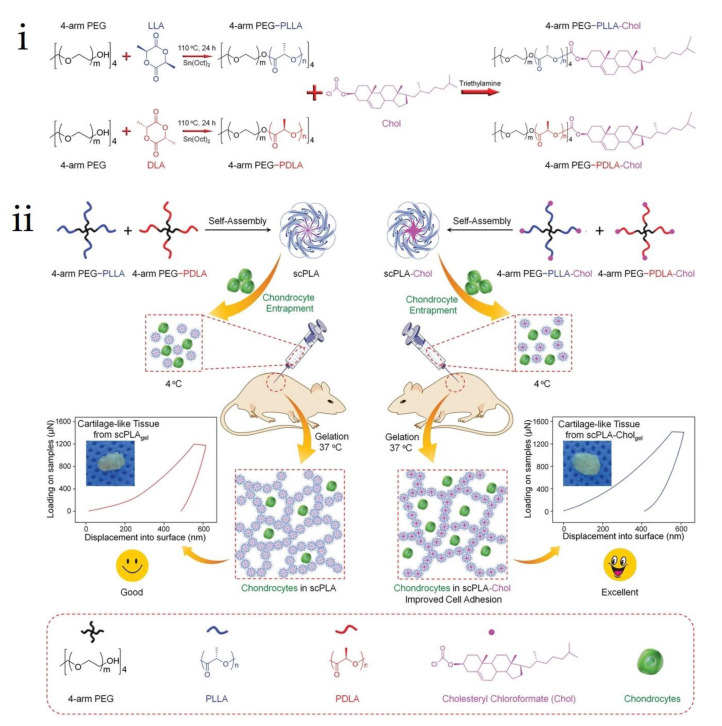
Schematic illustrations for (**i**) copolymer synthesis of 4-armPEG-PLL(D)A-cholesterol and (**ii**) in vivo cartilage regeneration of thermogels entrapped chondrocytes via subcutaneous injection into nude mice [86], Copyright 2019. Reproduced with permission from WILEY-VCH Verlag GmbH & Co.

**Figure 19 molecules-26-02846-f019:**
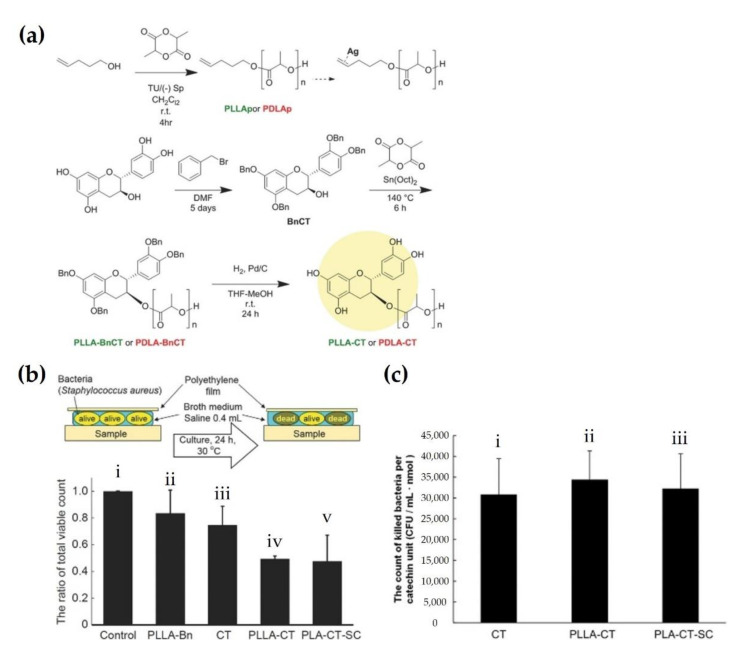
(**a**) Synthesis process schematic of chain-end modified PLLA and PDLA. (**b**) Comparison of antibacterial properties through ratio of live bacteria counts of (**i**) control, (**ii**) PLLA-Bn, (**iii**) CT, (iv) PLLA-CT, and (**v**) PLA-CT-SC against the control group. Top scheme indicates a method for measurement of antibacterial properties. The graphs were represented as mean values ± SD (*n* = 3). (**c**) Comparison of the counts of killed bacteria per CT unit of (**i**) CT, (ii) PLLA-CT, and (**iii**) PLA-CT-SC. The graphs were represented as mean values ± SD (*n* = 3) [102], Copyright 2016. Reproduced with permission from WILEY-VCH Verlag GmbH & Co.

## Data Availability

Not applicable.

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
