# Peer review of "Stereocomplex Polylactide for Drug Delivery and Biomedical Applications: A Review"

_molecules, 2021, doi:10.3390/molecules26102846_

Round 1

Reviewer 1 Report

Overall, the authors have achieved their goal in describing several cases where stereocomplexed poly-lactide has been investigated for drug delivery and bioapplications.  Several vignettes are presented that showcase examples of this.  One critical issue is the lack of pertinent references such as

  1. AUTHOR=Luo Fuhong, Fortenberry Alexander, Ren Jie, Qiang Zhe;TITLE=Recent Progress in Enhancing Poly(Lactic Acid) Stereocomplex Formation for Material Property Improvement; JOURNAL=Frontiers in Chemistry YEAR=2020; PAGES=688
  2. Bai H, Deng S, Bai D, Zhang Q, Fu Q. Recent Advances in Processing of Stereocomplex-Type Polylactide. Macromol Rapid Commun. 2017 Dec;38(23). doi: 10.1002/marc.201700454. Epub 2017 Sep 12. PMID: 28898498.
  3. Tsuji H. Poly(lactic acid) stereocomplexes: A decade of progress. Adv Drug Deliv Rev. 2016 Dec 15;107:97-135. doi: 10.1016/j.addr.2016.04.017. Epub 2016 Apr 25. PMID: 27125192.

The last reference is part of a large special issue in Advanced Drug Delivery Reviews: “PLA biodegradable polymers”; Edited by Abraham J Domb, Robert Langer, Arijit Basu; Volume 107, Pages 1-392 (15 December 2016).

All of the preceeding publications were not properly referenced, and topics in the current manuscript have been treated in these reviews.

Further comments are:

Line 153:  Is there supposed to be stereochemistry indicated for “whereas the diameter of mPEG45-PH30-PLA82”

Line 379:  The authors define “hydrogel, a network of cross-linked polymer chains, consists of 90 % water”; however, this is an arbitrary water amount.  A hydrogel can have varying amounts of water

Lines 389-390:  The authors state “Enantiomeric PLA oligomers grafted to dextran did not require artificial agents…”  What’s considered artificial?

Lines 425-430  The authors state “However, conventional hydrogels have limitations in various applications due to their poor mechanical properties and stability. In contrast, sc-PLA-based hydrogels have the potential to improve the mechanical strength and durability and delay the degradation rate of the carrier induced by stereocomplexation….”  This statement is made without references.  There are many examples showing the above comment is not true.

Lines 419-425:  The authors state “The strategy using in situ gelling systems has been used for the transformation of 419 drug/polymer precursor complexes from solution after injection into the human body to 420 gel form by physiological conditions of target tissues or artificial stimuli, such as pH or 421 temperature change, UV irradiation, solvent exchange, catalytic ions, or molecules [46-422 48]. Generally, an in situ gelling hydrogel can be synthesized by various chemical reac-423 tions, including enzyme-catalyzed cross-linking, Schiff-base reaction, photo-induced 424 polymerization, and Michael-type addition [49-52]. This drug delivery system can pre-425 vent adverse events in non-target tissues with improved availability of administration.”  However, it is not clear that the sc-PLA-based hydrogels described are reported for in situ results, or that scPLA has been formed in-situ.

Lines 635-636:  The authors state “Therefore, biomaterials based on sc-PLA are critical for securing anti-microbial effects to pre-636 vent bacterial proliferation to decrease adverse events and maximize clinical efficacy.”  This statement makes it sound like sc-PLA has antimicrobial effects; however this is not the case, all examples showed other components than sc-PLA are the anti-microbials.

Author Response

Reviewer: 1

We would like to sincerely thank the reviewer for the careful and thorough reading of the manuscript and for the thoughtful comments and constructive suggestions, which helping in greatly improving the quality of our manuscript.

Comments to the author(s):

  1. One critical issue is the lack of pertinent references such as
  1. AUTHOR=Luo Fuhong, Fortenberry Alexander, Ren Jie, Qiang Zhe;TITLE=Recent Progress in Enhancing Poly(Lactic Acid) Stereocomplex Formation for Material Property Improvement; JOURNAL=Frontiers in Chemistry YEAR=2020; PAGES=688
  2. Bai H, Deng S, Bai D, Zhang Q, Fu Q. Recent Advances in Processing of Stereocomplex-Type Polylactide. Macromol Rapid Commun. 2017 Dec;38(23). doi: 10.1002/marc.201700454. Epub 2017 Sep 12. PMID: 28898498.
  3. Tsuji H. Poly(lactic acid) stereocomplexes: A decade of progress. Adv Drug Deliv Rev. 2016 Dec 15;107:97-135. doi: 10.1016/j.addr.2016.04.017. Epub 2016 Apr 25. PMID: 27125192.

The last reference is part of a large special issue in Advanced Drug Delivery Reviews: “PLA biodegradable polymers”; Edited by Abraham J Domb, Robert Langer, Arijit Basu; Volume 107, Pages 1-392 (15 December 2016).

All of the preceeding publications were not properly referenced, and topics in the current manuscript have been treated in these reviews.

Reply to the comment : Firstly, we sincerely appreciate the reviewer for his valuable comment. Following your comment, we have added above references wherever necessary [10-12].

  1. Line 153:  Is there supposed to be stereochemistry indicated for “whereas the diameter of mPEG45-PH30-PLA82”

 Reply to the comment : Thank you for your valuable comment. The mean diameter of the stereocomplex nanoparticles slightly decreased when the pH changed from 5.0 to 7.9, as shown in Figure 3b and c. It was considered that lower pH conditions caused the swelling of the nanoparticles with the protonation of poly(L-histidine) in the tri-block copolymer.

  1. Line 379:  The authors define “hydrogel, a network of cross-linked polymer chains, consists of 90 % water”; however, this is an arbitrary water amount.  A hydrogel can have varying amounts of water

 Reply to the comment : Thank you for your constructive comment. Following your comment, we have removed the misleading phrase “consists of 90 % water” in line 379.

  1. Lines 389-390:  The authors state “Enantiomeric PLA oligomers grafted to dextran did not require artificial agents…”  What’s considered artificial?

 Reply to the comment : The “artificial agents” means not natural or chemically produced agents, such as crosslinking agents and nucleating agents.

  1. Lines 425-430  The authors state “However, conventional hydrogels have limitations in various applications due to their poor mechanical properties and stability. In contrast, sc-PLA-based hydrogels have the potential to improve the mechanical strength and durability and delay the degradation rate of the carrier induced by stereocomplexation….”  This statement is made without references.  There are many examples showing the above comment is not true.

 Reply to the comment : Thank you for your valuable comment. We agree with your opinion. Thus, we have removed the misleading sentence “However, conventional hydrogels have limitations in various applications due to their poor mechanical properties and stability.” In line 426.

  1. Lines 419-425:  The authors state “The strategy using in situ gelling systems has been used for the transformation of 419 drug/polymer precursor complexes from solution after injection into the human body to 420 gel form by physiological conditions of target tissues or artificial stimuli, such as pH or 421 temperature change, UV irradiation, solvent exchange, catalytic ions, or molecules [46-422 48]. Generally, an in situ gelling hydrogel can be synthesized by various chemical reac-423 tions, including enzyme-catalyzed cross-linking, Schiff-base reaction, photo-induced 424 polymerization, and Michael-type addition [49-52]. This drug delivery system can pre-425 vent adverse events in non-target tissues with improved availability of administration.”  However, it is not clear that the sc-PLA-based hydrogels described are reported for in situ results, or that scPLA has been formed in-situ.

Reply to the comment : Firstly, we are sorry to confuse your understanding. The strategy using in situ gelling has never been tried until now. Thus, we suggested that in situ gelling system can be applied to sc-PLA-based hydrogels in the future, because it has many advantages such as target specificity and improved availability of administration. As your good comment, the description can cause misunderstanding to readers. Therefore, we have revised the sentence as follows: “Based on this strategy, sc-PLA-based hydrogels have the potential to improve the mechanical strength and durability and delay the degradation rate of the carrier induced by stereocomplexation in the future.” in line 426.

  1. Lines 635-636:  The authors state “Therefore, biomaterials based on sc-PLA are critical for securing anti-microbial effects to pre-636 vent bacterial proliferation to decrease adverse events and maximize clinical efficacy.”  This statement makes it sound like sc-PLA has antimicrobial effects; however this is not the case, all examples showed other components than sc-PLA are the anti-microbials.

Reply to the comment : Thank you for your constructive comment. As you mentioned, neat sc-PLA cannot exhibit antimicrobial effects. In fact, the sentence means that biomaterials based on sc-PLA are necessary to obtain anti-microbial effects because it does not have innate antibacterial effect.

Reviewer 2 Report

The manuscript is " Stereocomplex Polylactide for Drug Delivery and Biomedical Applications: A Review ".

General comments:

PLA is one of the most used biodegradable polymer in various fields. Authors introduce the sc-PLA for drug delivery, biomedical applications based from previous studies. However, authors did not put forward their own innovative thesis, and the future applicable fields. Therefore, I think authors should clearly confirm the synthesis mechanism and characteristics of different sc-PLAs.

Author Response

Point-by-point replies to the referee comments

Reviewer: 2

We would like to sincerely thank the reviewer for the careful and thorough reading of the manuscript and for the thoughtful comments and constructive suggestions, which helped in greatly improving the quality of our manuscript.

Comments to the author(s):

PLA is one of the most used biodegradable polymer in various fields. Authors introduce the sc-PLA for drug delivery, biomedical applications based from previous studies. However, authors did not put forward their own innovative thesis, and the future applicable fields. Therefore, I think authors should clearly confirm the synthesis mechanism and characteristics of different sc-PLAs.

Reply to the comment : We sincerely appreciate the reviewer for the valuable suggestion. As your suggestion, we described future applicable fields and perspectives of sc-PLA in a Conclusions part as following; “3D printing technology has typically used PLA as filament material; however, sc-PLA has not been used in industrial applications to date. As industrial application fields and potential of 3D-printing extended, inkjet printing using sc-PLA with improved characteristics proves a versatile technology to simultaneously induce both stereocomplexation and fabrication in the future, if inkjet printing is suitably converged with 3D printing techniques. In biomedical applications, sc-PLA nanoparticles are potential carriers for anti-cancer therapy and scaffolds for tissue engineering. Moreover, numerous studies have demonstrated that they can be improved by diverse methods, such as surface modification and co-polymerization, and attain additional functions, such as anti-microbial effects and immune stealth. sc-PLA nanoparticles effectively encapsulate therapeutic anti-tumor agents, such as DOX, and specifically transfer the agents to target lesions.”

Reviewer 3 Report

The review focuses on the recent advances in research fields involving the use of sc-PLA in great detail. The chemical versatility of such biomaterials is high and it is largerly demonstrated by the way in which the authors refer to the materials processing for application as micelles, self-assembly, emulsion, and 3D- printing. The authors well evidenced that this biomaterials is expected to become a next-generation polymeric material owing its physico-chemical properties

Author Response

Point-by-point replies to the referee comments

Reviewer: 3

We would like to sincerely thank the reviewer for the careful and thorough reading of the manuscript and for the thoughtful comments and constructive suggestions, which helped in greatly improving the quality of our manuscript.

Comments to the author(s):

The review focuses on the recent advances in research fields involving the use of sc-PLA in great detail. The chemical versatility of such biomaterials is high and it is largerly demonstrated by the way in which the authors refer to the materials processing for application as micelles, self-assembly, emulsion, and 3D- printing. The authors well evidenced that this biomaterials is expected to become a next-generation polymeric material owing its physico-chemical properties.

Reply to the comment : We sincerely appreciate the reviewer for the generous and encouraging comment.